# Immunity in Sea Turtles: Review of a Host-Pathogen Arms Race Millions of Years in the Running

**DOI:** 10.3390/ani13040556

**Published:** 2023-02-05

**Authors:** Alana Nash, Elizabeth J. Ryan

**Affiliations:** 1Department of Biological Sciences, Faculty of Science and Engineering, University of Limerick, V94 T9PX Limerick, Ireland; 2Health Research Institute, University of Limerick, V94 T9PX Limerick, Ireland; 3Limerick Digital Cancer Research Centre, University of Limerick, V94 T9PX Limerick, Ireland

**Keywords:** eco-immunology, reptile, sea-turtle, fibropapillomatosis, innate and adaptive immunity

## Abstract

**Simple Summary:**

Sea Turtles have a unique immune system, which evolved over millions of years in the persistent host-pathogen arms race. As this species occupies a unique evolutionary and environmental niche, they provide an opportunity to gain insight into the evolution of immunity. We present an overview of the turtle immune system, including the cells and organs important for coordinating the immune response to pathogens, with a focus on pathogen recognition and inflammatory mediators, including Interferons. We highlight areas for future study and note which studies have investigated freshwater turtles and are lacking in sea turtles. We particularly focus on the Green Sea turtle (*Chelonia mydas*) as the juvenile turtles within this species are the most afflicted by the neoplastic tumorous disease, fibropapillomatosis (FP).

**Abstract:**

The immune system of sea turtles is not completely understood. Sea turtles (as reptiles) bridge a unique evolutionary gap, being ectothermic vertebrates like fish and amphibians and amniotes like birds and mammals. Turtles are ectotherms; thus, their immune system is influenced by environmental conditions like temperature and season. We aim to review the turtle immune system and note what studies have investigated sea turtles and the effect of the environment on the immune response. Turtles rely heavily on the nonspecific innate response rather than the specific adaptive response. Turtles’ innate immune effectors include antimicrobial peptides, complement, and nonspecific leukocytes. The antiviral defense is understudied in terms of the diversity of pathogen receptors and interferon function. Turtles also mount adaptive responses to pathogens. Lymphoid structures responsible for lymphocyte activation and maturation are either missing in reptiles or function is affected by season. Turtles are a marker of health for their marine environment, and their immune system is commonly dysregulated because of disease or contaminants. Fibropapillomatosis (FP) is a tumorous disease that afflicts sea turtles and is thought to be caused by a virus and an environmental factor. We aim, by exploring the current understanding of the immune system in turtles, to aid the investigation of environmental factors that contribute to the pathogenesis of this disease and provide options for immunotherapy.

## 1. Introduction

The immune system has evolved to protect the host in the constant ‘host-pathogen arms race’. Pathogens evolve mechanisms to evade immunity, and the immune system must counterattack [1]. Our understanding of immunology is enriched by the study of a broad spectrum of species that encounter a diversity of microbial organisms. Many fundamental immunology concepts have been discovered through the study of non-mammalian or non-model organisms such as drosophila (discovery of Toll-like receptors [2]), starfish and sea urchins (concept of self-versus nonself recognition [3]), and chickens (discovery of T and B cell lineages and genetic mechanism of antibody diversity [4,5]). As sea turtles are long-lived ectotherms with a unique evolutionary history, further study of this ancient immune system could provide further insight into fundamental immunology that could lead to the development of new therapeutic approaches for a variety of diseases and potentially discover new antimicrobial effectors [6,7].

Out of the seven surviving sea turtle species, three are classed as ‘Vulnerable’; Loggerhead (*Caretta caretta* (Linnaeus, 1758)), Leatherback (*Dermochelys coriacea* (Vandelli, 1761)), and Olive Ridley (*Lepidochelys olivacea*, (Eschscholtz, 1829), one as ‘Endangered’; Green (*Chelonia mydas* (Linnaeus, 1758)), two as ‘Critically Endangered’; Kemp’s Ridley (*Lepidochelys kempii* (Garman, 1880)) and Hawksbill (*Eretmochelys imbricata* (Linnaeus, 1766)), and the flatback (*Natator depressus* (Garman, 1880)) is ‘data deficient’ according to the IUCN (International Union for Conservation of Nature and Natural Resources) [8]. The geographic distribution and phylogenetic relationships of the turtle species discussed in this review are outlined in Table 1 and Figure 1. Sea turtles live in a unique hyper-saline and microbe-laden marine environment. As a result, Sea turtles’ immune response has evolved its own unique approach to handling the highly conserved functions of host defence. Understanding how their environment has influenced variation in immune genes and regulation of immunity is important in developing long-term strategies to protect sea turtles from both existing and emerging infectious diseases.

The immune response of Testudines (turtles, terrapins, and tortoises) is the most studied of the surviving lineages in class Reptilia, which includes Crocodilia (crocodilians), Squamata (snakes, lizards, amphisbaenians) and Rhychocephalia (tuatara) [65]. Yet, these studies represent only 12% of the 365 species of Testudines [65]. Studies have focused on turtles’ heavy reliance upon the robust innate response to pathogens, leaving questions regarding the role of the adaptive response. The anti-viral response, in particular, is poorly understood [19,66,67]. Turtles’ lymphocytes are believed to respond to infection in a relatively nonspecific way with an altered antibody and memory response [66]. Large gaps in knowledge of the sea turtle immune system include; the mechanism of pathogen recognition through pattern recognition receptors (PRRs), the pathways and efficiency of antiviral effectors such as interferons (IFNs) and IFN-stimulated genes (ISGs), the role of cytotoxic T cells in killing infected or damaged host cells, and the mechanism and locations of T cell-mediated B cell activation [66].

Understanding these gaps will not only elucidate the evolution and diversity of the immune system but will also assist in the conservation of vulnerable sea turtle species from threats of chronic disease, infection, and a degrading marine environment. For example, fibropapillomatosis (FP) is a tumorous disease that affects sea turtles. The dramatic increase in FP prevalence since the 1980s is attributed to a virus, and an unknown environmental factor thought to be anthropogenic in nature [68,69,70]. Results from genomic [71], proteomic [72,73,74] and transcriptomic studies [70,75,76] have consistently identified immune genes as dysregulated between healthy and diseased turtles or by levels of exposure to marine contaminants [73]. Additionally, sea turtles are possible reservoirs for zoonotic viruses, as *C. mydas*, along with snakes and pangolins, have been suggested as potential intermediate hosts of SARS-CoV-2 [77]. Incidences of antibiotic-resistant bacterial infections in sea turtles have been increasing as antibiotics are continually released into the environment [34,78,79,80,81,82,83,84].

Multidisciplinary interest in sea turtle immunology is growing, guided by ecoimmunology and ‘One Health’ principles, and the application of omics techniques, protein isolation, and purification, microscopy, and histology will aid in the characterization of immune components [65,67,85]. The adaption, standardization, and application of these techniques to sea turtle research will continue to identify dysregulated and disease-related pathways, allowing the selection of potential precision medicine treatments [70,85]. However, the turtle immune response is influenced by individual and species-specific factors, including season, sex, and temperature [19,33,86,87]. Thus, further fundamental understanding and characterization of the immune response is required to achieve future successful therapies.

Studies of sea turtle immune function have been focused on the protection and conservation of turtle wildlife from threats such as FP, parasites, or environmental contaminants [42,49,50,51,52,53,54,55,56,73,75,76,88,89,90,91,92]. Functional studies of the turtle immune system have been carried out on red-eared slider turtles *Trachemys scripta* (Thunberg, 1792)) [19,20,21,22,23], Caspian turtle or striped-neck terrapin (*Mauremys capsica* (Gmelin, 1774)) [27,28,29,30,31] and Chinese soft-shelled turtle (*Pelodiscus sinensis* (Wiegmann, 1853)) [14,87,87,93]. In this review, we discuss the innate and adaptive immune systems of sea turtles, and we report what studies have examined the effects of the environment on the sea turtle immune function. We focus particularly on *C. mydas*. This is because, as juveniles, these turtles are most frequently afflicted by FP. Furthermore, the *C. mydas* genome was the first published and thus most investigated sea turtle genome.

## 2. Lymphoid Structures

Reptile lymphoid structures offer a first glance at the distinctiveness of their immune system compared to mammals. Missing or altered lymphoid structures highlight how the reptilian immune system is biased to the evolutionarily older innate response as a robust defence against infection [86]. Furthermore, seasonal changes in lymphoid structures provide evidence for an altered reliance upon the adaptive response [19,28,30,86].

Lymphoid structures are organs in which immune effector cells originate, mature, detect antigens, and induce an immune response. Primary lymphoid structures in the turtle are the bone marrow and thymus (Figure 2) [86]. Bone marrow is the site of haematopoiesis, the production of red and white blood cells (erythrocytes and leukocytes) and plasma in blood. Leukocytes differentiate into innate effectors from myeloid progenitor cells (heterophils, eosinophils, basophils, monocytes) or adaptive effectors from lymphoid progenitor cells (T and B cells). (Natural killer (NK) cells are an exception as they are innate lymphocytes). As in mammals, turtle T cells migrate and mature in the thymus [59,63]. B cells migrate to the spleen, which is the largest antibody-forming organ in turtles. The spleen has been described in a snapping turtle (*Chelydra serpentina*, Linnaeus, 1758) [63], the Caspian turtle (*M. capsica*) [28,29] and the Chinese soft-shelled turtle (*P. sinensis*) [9].

Other than the spleen, secondary lymphoid structures in turtles include gut-associated lymphoid tissue (GALT) and isolated lymphoid follicle-like structures (ILFs) (Figure 2). The GALT and ILFs concentrate antigens to interact with lymphocytes in tissue and induce an immune response. Murine ILFs are organized lymphoid structures found in the GALT that contain lymphocytes, predominately antibody-producing B-2 subtype B cells [94]. ILFs have a unique inducible nature, and changes in the local microbial community influence distribution [94,95]. B cells containing ILF-like aggregates were discovered in the intestine of the red-eared slider turtle (*T. scripta*) [21] and in the digestive tract, urinary bladder, cloaca, larynx, and trachea of *C. mydas* [59].

Reptiles lack both lymph nodes and Peyer’s patches, both of which are key sites for the initiation of the mammalian adaptive response (Figure 2). Lymph nodes are areas of high lymphocyte density where lymphatic fluid is filtered, and antigens are presented to lymphocytes. Germinal centers are found here and are the site of B lymphocyte maturation [96]. ILFs are thought to represent the ancestor of mammalian lymph nodes and replace the missing function of lymph nodes [63,86]. Peyer’s patches are located in the GALT and consist of isolated or aggregated lymphoid follicles with a function to induce immune tolerance or defense against pathogens by the interplay between the immune cells and the follicle-associated epithelium [97]. ILFs are suggested to trap and process antigens for presentation to cells of the immune response, are possible sites of B cell maturation, and play a role in mucosal immunity [21,59]. The mechanisms regulating immune cell trafficking to and from ILFs and the cell: cell interactions within the ILFs required for optimal immunity have not been fully described.

### Seasonal Changes of Lymphoid Structures

The structure of the reptilian spleen and thymus is season-dependent [86]. Seasonal involution of the thymus and spleen results from neuroendocrine interactions that occur in all ectothermic vertebrates. This has been studied in lizards, snakes, and freshwater turtles [28,30,98,99]. The key effector cells of the adaptive immune system, T and B cells, mature and are activated in ILFs. Thus, reptiles cannot rely on adaptive immunity in the same manner as mammals all year round [86]. The precise patterns of seasonal variations in the structure are species-specific [86]. To our knowledge, this effect is yet to be discussed, specifically in sea turtles. However, a study of the terrapin *Mauremys capsica* shows that the spleen and thymus undergo seasonal variation affecting lymphocyte density and differentiation [28,30]. As Winter gives way to Spring, the thymus gland increases in size, and the cortex begins to develop. By the end of Spring, the thymus reached its maximum size with a well-developed medulla and cortex, and adaptive lymphocytes are most abundant. In Summer, the thymus involutes greatly, and lymphocyte density declines to its lowest in Autumn. The thymus undergoes a small increase in Autumn, only to gradually decrease again during Winter [28,30]. Innate immune effectors, heterophils, basophils, and monocytes, reach their highest levels in Autumn [30]. In addition, Muñoz et al. reported a sex difference in the distribution of lymphocytes; females had higher proportions of lymphocytes than males in the thymus during Summer and in the spleen during Spring [30]. Variation in lymphoid structure implies that turtles rely more heavily upon the innate immune system in Autumn and Winter. Adaptive immunity bolsters defense against infection during Spring providing enhanced protection during mating and nesting [19].

## 3. Innate Immune System

Reptiles rely on their robust innate immune system to protect themselves from infection. The innate immune system is an organism’s first line of defense against invading pathogens. It is non-specific and needs no previous exposure to a pathogen. The first barrier against microbial invasion is a turtle’s thick, keratinized outer layer of skin [100]. This antimicrobial epidermal barrier is surveilled by Langerhans cells and is aided by antimicrobial peptides (AMPs) [101]. If a pathogen succeeds in breaching this barrier, the innate immune system mobilizes a broad spectrum of non-specific responses, including inflammation, behavioral fever, innate effectors, and non-specific leukocytes, and activates the adaptive response. [67]. Inflammatory mediators signal the recruitment of heterophils, not neutrophils, as in mammals, to the site of infection and begin the immune response [86]. In this section, we discuss what is known about the innate immune system in the turtle (summarized in Figure 3).

### Innate Effector Cells

Although reptiles and mammals as vertebrates have similarities in innate immune genes, there are frequent differences in their function [67]. As interest in the field of eco-immunology grows and the list of published reptilian genomes gets longer, comparative studies reveal the diverse evolution and mechanisms of host defense [102,103,104]. In reptiles, the dividing line between innate and adaptive immunity is blurred. For example, adaptive cells take part in traditional innate functions like phagocytosis. Even in mammals, innate effector cells undergo ‘trained immunity’ where the second exposure to an antigen can induce a stronger response which is the traditional adaptive characteristic of memory. (Trained immunity is the heritable epigenetic changes, first described as in monocytes and macrophages, that induce a state of hyper- or hypo-responsiveness to future infection [105,106,107,108,109]). Turtle innate effector cells consist of non-specific leukocytes: monocytes, macrophages, dendritic cells, NK cells, heterophils, eosinophils, and basophils [110,111].

Phagocytosis is an extremely conserved function across many species and is one of the fundamental functions of innate immune effector cells. After phagocytosis of a pathogen, a phagocyte releases cytokines to modulate the immune response, and professional antigen-presenting cells can initiate the adaptive response through the activation of T cells. Turtles have a wide variety of phagocytes, including monocytes, macrophages, melanomacrophages, heterophils, eosinophils, thrombocytes, and B cells [89,110,111,112]. The major phagocytic effectors of the innate immune response in sea turtles are reported to be heterophils and lymphocytes [111,113]. *C. mydas* lymphocytes were identified as major phagocytes, although it is unclear if this includes both T and B cells [111].

Turtle granulocytes include heterophils, eosinophils, and basophils. Heterophils are large round granulocytes with an unlobed nucleus, thought to replace the function of mammalian neutrophils. These cells assist in suppressing microbial invasion by the release of antimicrobial peptides and are involved in the inflammatory response [86]. Eosinophils are larger in sea turtles compared to freshwater turtles [114]. They are phagocytic cells involved in the removal of parasitic pathogens. High numbers in blood are associated with innate immune stimulation and parasitic infection [114]. A study in snapping turtles reported eosinophils could phagocytize immune complexes [62]. Basophils are one of the only leukocytes in turtles, not yet described to have phagocytic properties. Basophils have variable morphology in sea turtles but are usually small in size with a centrally located nucleus [114]. Basophils have a parallel immune function to that of mammalian basophils or mast cells, both containing 5-hydroxytryptamine and histamine. Basophil cell surface contains antigen-specific immunoglobulins that, when activated, induce degranulation and release of histamine [114]. The level of histamine released increases as the concentration of antigen increases [115]. This reaction time in turtles lasts significantly longer (40–60 min) in comparison to humans (2 min) [64].

Monocytes can differentiate into macrophages and dendritic cells [116]. Turtle monocytes have the most variable morphology of the leukocytes and can be rhomboid, oval, or round in shape and contain basophilic cytoplasm [114]. Monocytes are phagocytic, antigen-presenting cells (APCs) and are involved with inflammatory disease through the release of cytokines [19]. Generally, monocytes are found at a low count in *C. mydas* [114]. However, count increases during chronic inflammation, bacterial or parasitic diseases, and chronic antigenic stimulation [113]. Langerhans cells are a dendritic cell subtype particularly found in the human epidermis and mucosal surfaces, and were reported in *C. mydas* epidermis [101]. These Langerin+ cells adopted a pleomorphic morphology and were observed around ulcerative dermatitis ulcers.

Respiratory burst is an ancient method of killing ingested microorganisms by toxic molecules observed in birds, reptiles, and fish [86,113]. After phagocytosis of the pathogen, phagocytes NADPH generate intracellular reactive oxygen species (ROS) via NADPH oxidase, which eliminates microorganisms from the host. Antioxidants are expressed in turtle blood as a defense against the physiological stress caused by excessive ROS production. Lymphocytes are suggested to be the major effectors of respiratory burst in loggerhead sea turtles, *C. caretta* [113].

## 4. Behaviorally Induced Fever

Fever is an integral part of the innate immune response in homeotherms. As ectotherms, turtles cannot self-regulate their body temperature. Instead, they increase their body temperature in response to infection by moving to warmer waters or basking (Figure 3) [86]. Behavioral fever activity raises body temperature by approximately 4 °C in response to infection or handling stress in a freshwater turtle (*Chrysemys picta*, Schneider, 1783), two terrestrial turtles (*Terrapene carolina carolina* (Linnaeus, 1758) and *Clemmys insculpta* (Le Conte, 1830)), and a tortoise (*Gopherus polyphemus*, Daudin, 1801) [18,117,118]. Temperature fluctuation can have a direct effect on innate immunity. Turtles’ innate effector cell function is influenced by temperature [64,86,119]. Snapping turtle basophils released histamine in a range of 10–37 °C, with an optimum of 27 °C [19,64]. In pond slider turtle (*T. scripta*), the macrophage-mediated respiratory burst was optimum between 25–30 °C [119].

Certain *C. mydas* populations in the Pacific Ocean, including Hawaii, engage in terrestrial basking behavior, due to cold winter sea surface temperature [120]. A 2006 study observed a relationship between FP and basking behavior in *C. mydas*. Only turtles with FP were observed basking, resulting in a 2.9 °C increase above ambient body temperature [121]. Heat-seeking behavior has been reported in murine cancer models, and cold stress is associated with accelerated tumor growth [122,123]. Further study is needed to define the relationship between infection and mechanisms governing behavioral fever in sea turtles and to understand if warming oceans will impact the fever response and how this will affect the immune response to infection and the incidence of cancer.

## 5. Innate Immune Effectors

Studies of reptilian innate immune effectors, including lysozyme, complement, and antimicrobial peptides, suggest these molecules have a broader range of activity in comparison to their mammalian counterparts (Figure 3) [115].

(1)Lysozyme

Lysozymes are enzymes that kill bacteria by catalyzing the hydrolysis of the cell wall [124]. Lysozymes are released by phagocytes into mucus, saliva, and plasma in response to foreign microorganisms. Reptilian lysozyme activity can vary on water temperature, pH, toxicants, sex, age, season, and stressors [124].

Lysozyme C (chicken) and G (goose) genes have been identified in the *C. mydas* genome [125]. Lysozyme C is the major lysozyme type, ubiquitously and constitutively expressed in vertebrates and all tissue types, respectively [126]. Lysozyme G has been described in mammals, birds, fish, and insects with activity against Gram-positive and Gram-negative bacteria. Lysozyme G is not constitutively expressed and is highly upregulated in response to infection. In fish, lysozyme G expression is upregulated after challenge with lipopolysaccharide (LPS), *Vibrio alginolyticus*, *Aermonas hydrophila*, and *Aphanomyces invadans* [126,127,128,129,130,131,132].

*C. mydas* lysozyme C has a dual pH optima at pH 6.0 and pH 8.0, contrasting with single pH optima of pH 6.0 in lysozyme isolated from soft-shelled turtles [133]. *C. mydas* lysozyme C had high activity against the Gram-positive bacteria; *Micrococcus luteus*, moderate activity to two Gram-positive and two Gram-negative bacteria; *Staphylococcus aureus* and *Staphylococcus epidermidis*, and *Pseudomonas aeruginosa* and *Vibrio cholerae*, respectively. Gram-negative bacteria *Pseudomonas* spp. and *Vibrio* spp. are associated with inflammatory diseases, including ulcerative dermatitis and ulcerative stomatitis. Interestingly, *C. mydas* lysozyme C showed mild activity against Gram-negative bacteria that are identified as hazardous bacterial threats to sea turtles. *Escherichia coli* and *Salmonella typhi* are opportunistic pathogens with zoonotic potential, while *Klebsiella pneumoniae* has been associated with regional mass mortalities. Turtles are vulnerable to disease when the innate response is inadequate to kill these pathogens.

Plasma lysozyme levels can act as an innate immune response biomarker and can help indicate the general state of health of a turtle [113]. Lysozyme activity in turtles decreases upon exposure to plastic pollution [35], heavy metals [36], fluctuating temperature [87], brevotoxins [22,37,53,134], and organochlorine contaminants [135]. Conversely, *P. sinensis* lysozyme activity is enhanced by increased dietary protein intake [136], linking good nutrition to immunity.

Lysozyme plays a role in the protection of turtle eggs from infection. Egg-white lysozyme has been characterized in *C. mydas* [133,137], *L. olivacea* [47], and soft-shelled turtles *P. sinensis* and *T. gangeticus* [138,139]. Interestingly, the egg white of loggerheads (*C. caretta*) is reported to lack lysozyme, and its antibacterial activity is suggested to be replaced by a small cationic β-defensin-like protein named TEWP (turtle egg white protein) [140].

(2)Antimicrobial Peptides (AMPs)

Antimicrobial peptides (AMPs), also known as host defense peptides (HDPs), in turtles include defensins and cathelicidins. Interestingly, turtle genomes encode ‘avian-like’ defensin-type peptides, yet their cathelicidins are ‘snake-like’ [125].

Defensins are one of the major classes of cationic antimicrobial peptides and are expressed in response to bacterial or viral infection [141]. Defensins are investigated for their powerful antimicrobial and immunomodulatory properties as a potential therapy against antibiotic-resistant bacteria. Defensins can modulate autophagy, apoptosis, pro- and anti-inflammation responses, wound healing, chemoattraction, and cellular differentiation [141,142]. Defensins are divided into three main sub-classes; α, β and θ. Turtles express β-defensins, humans express both α and β, while primates express θ [125]. Three defensins have been described thus far in turtles, turtle β-defensin 1 (TBD-1), pelovaterin, and turtle egg white protein (TEWP). The antimicrobial activity of some turtle defensins has been investigated, although their mechanisms of immune modulation are not yet characterized to our knowledge.

A 40-residue peptide, called TBD-1, was the first defensin isolated from reptilian leukocytes in 2008 from a European pond turtle (*Emys orbicularis*, Linnaeus, 1758) [143]. TBD-1 showed high antimicrobial activity against *E. coli*, *L. monocytogenes* and low activity against *S. aureus* and *C. albicans*, a similar antimicrobial activity as the control human and pig defensins included in the study [143]. TBD-1 has activity against both gram-positive and negative bacteria and has antifungal activity. Turtle eggshell contains antimicrobial peptides. Pelovaterin is the major intracrystalline component in Chinese soft-shelled (*P. sinensis*) eggshells. Pelovaterin has a similar tertiary structure to mammalian β-defensins, yet very low sequence identity and a distinctive anionic charge. Lakshminarayanan et al. demonstrated pelovaterins strong antimicrobial activity against two gram-negative bacteria *Pseudomonas aeruginosa* and *Proteus vulgaris*, moderate activity against gram-positive and *Staphylococcus* aureus and poor activity against gram-negative *E. coli* and *Enterobacter aerogenes*. [93]. TEWP is an ovodensin, a cysteine-rich AMP abundantly found in the egg white of reptiles and birds with varying degrees of antimicrobial activity. Vertebrate defensins are characterized by the presence of a conserved six-cysteine motif forming three disulfide bridges. However, chicken and *C. mydas* ovodefensins exhibit different disulfide bridge patterns [140,144,145]. TEWP is present in *C. caretta* egg white and displays anti-microbial activity against *E. coli*, *Salmonella* spp., and Chandipura virus. As previously mentioned, the loggerhead turtle egg white lacks lysozyme, TEWP plays a particularly important role in the protection of these eggs from infection [140].

Further understanding of the diverse role of turtle defensins in protection from infection and disease could have therapeutic potential. Defensins are investigated for both pro- and anti-tumor properties [146,147]. In mammals, Ultraviolet (UV) exposure can stimulate the innate response and suppress the adaptive response [148,149]. UV-B exposure increased the expression of human defensins, including β-defensins, from keratinocytes [149]. In *C. mydas*, UV exposure improved FP prognosis and increased vitamin D levels [56]. However, UV exposure in humans is characteristically known for its immunosuppressive qualities on the adaptive response [148]. As turtles rely on their innate response, the study of sea turtle defensins and how the expression is influenced by the environment may have therapeutic potential for prevalent diseases or as a method to boost natural immunity against antibiotic-resistant bacteria [150].

Cathelicidins are another major class of cationic antimicrobial peptides with broad-spectrum activity against bacteria, fungi, and enveloped viruses and the ability to trigger the innate immune response [151]. Four cathelicidins have been described in *C. mydas* and six in *P. sinensis* [7,13]. *C. mydas* cathelicidins (Cm-CATH1-4) are both constitutively expressed and inducible in response to in vivo bacterial challenges [7]. Cm-CATH2 can directly kill pathogens via membrane permeabilization, and other family members boost the immune response by activation of MAPKs and NF-kB-dependent signaling pathways. *C. mydas* cathelicidins have been suggested as a candidate for novel antimicrobial drug development due to their potent, broad spectrum, and rapid bactericidal and anti-biofilm activities. Cm-CATH2 boosted immune response by inducing the trafficking of nonspecific leukocytes to the infection site. However, CmCATH2 could also block the toll-like receptor (TLR) complex to prevent the LPS-mediated induction of proinflammatory cytokines in an in vitro cell culture assay [7]. While this finding should be validated in vivo, this study suggests that this class of AMP has a range of anti-microbial and immunomodulatory properties in sea turtles that merit further study.

(3)Complement System

The complement system has potent anti-microbial and pro-inflammatory properties [86,152]. The complement cascade involves a series of proteins in plasma that, when activated, can lead to the formation of the membrane attack complex (MAC) that causes lysis when inserted into microbial cell walls. In addition, complement activation initiates the recruitment of leukocytes to the site of infection, phagocytosis, and expression of pro-inflammatory cytokines (Figure 3) [61,152].

All three complement pathways, classical, alternative and lectin, have been observed in reptiles [67]. The classical pathway is a link between innate and adaptive immunity by involving the recognition and binding of antigens by antibodies [61]. The alternative pathway is activated in response to viral or bacterial particles such as LPS. Mannose residues on the surface of bacteria activate the lectin pathway [152]. Only in recent years have all three complement pathways been reported as being a part of the turtle immune response to bacterial challenge. Key immune response genes, including C1q (classic), MASP2 (lectin), CFB, and CFD (alternative), and complement C3 attack complex members were significantly upregulated in turtle liver after *Edwardsiella tarda* infection [15]. Complement C3 mRNA levels increased in soft-shelled turtle liver after infection with *Aeromonas hydrophilia* in a similar manner to that reported in humans, mice, and zebrafish [153,154].

Turtle complement is activated in response to environmental stressors, including the plastic-derived contaminant PBDE-47 [35] and nitrite [10]. At the same time, complement activity is suppressed in response to temperature fluctuation [27,155,156], the pesticide diazinon, and the herbicide atrazine [24,31]. Complement proteins have an optimal temperature range, and complement cascade proteins (C3, C4, C6-C9, and MASP1/2) expression was significantly downregulated at 32 °C compared to 28 °C in the pond turtle *M. mutica* (Cantor, 1842) [27]. In an investigation of the effects of different sunlamp-based lighting on captive soft-shelled turtles, turtles exposed to light in the basking area showed a significant increase in C3 in comparison to those without light exposure. These results demonstrate how turtles respond to temperature and pollutants in their environment by modulation of the complement cascade [157].

Similar turtle species rely on their innate immune response in an individual and species-specific manner. A study of the complement activity of two species of snapping turtles, the Common Snapping Turtle (CST, *C. serpentina*) and the Alligator Snapping Turtle (AST, *Macrochelys temminckii*, Troost, 1835), showed that CST relied on the alternative pathway. However, the AST relied upon the Lectin-activated pathway [61]. In a similar experiment comparing eastern (EBT) and ornate (OBT) box turtles, EBT relied on the lectin pathway and OBD, the alternative pathway [17]. Both turtles display bactericidal capacity. However, OBT displayed greater bactericidal capacity at a broader range (20–40 °C) against *Salmonella typhimurium*, *Escherichia coli*, *Enterobacter cloacae*, *Citrobacter freundi*, *Bacillus subtilis*, *Staphylococcus epidermidis*, and *Staphylococcus aureus* [17]. The results are unique as serum complement pathways are assumed to be conserved within clades and highlight the need for avoiding the generalization of immune functions between turtle species [61]. Regarding sea turtles, complement proteins (C3, C4, C5) have been highlighted as a potential biomarker of *C. caretta* [38] and *C. mydas* [73] health when recovering in rehabilitation facilities as they serve as good indicators of stress, infection, or exposure to pollutants.

## 6. Natural Abs

Natural antibodies (NAbs) are a broad first line of defense and are mediators between innate and adaptive responses. They are germline-encoded, polyreactive to PAMPs, and have a low-binding affinity [86]. NAbs can stimulate both the innate and adaptive immune system through the activation of complement and specific lymphocytes [86]. Reptilian humoral responses are slower and longer lasting than those of mammals. Following immunization, both mammals and reptiles have detectable antibody levels after about one week. In mammals, peak antibody production is after two weeks, while in reptiles, the production peaks between 6–8 weeks post-immunization [115]. However, reptiles’ responses are long-lasting responses with antibodies detectable up to 34 weeks post-immunization [115]. It is suggested that the longer-lasting NAb response in reptiles is a ‘trade-off’ when T cell proliferation is diminished due to seasonal changes of the spleen and thymus [67,158].

Mucosal NAbs in the red-eared slider turtles (*Trachemys scripta*) are non-specific, respond equivalently to new and previously exposed antigens, and are thought to be of the IgM subset produced by B-1 cells [19,159]. NAbs agglutination was not affected by exposure to the harmful algae toxin microcystin in the painted turtle (*Chrysemys picta*) [160]. NAbs are thought to play a key role in the robustness of the reptilian innate response. The B1 subset has a long half-life, is expressed throughout plasma and mucus, and can mount a specific and non-specific immune response. This is thought to protect long-lived turtles as levels of NAbs in mucus increases with age [159]. Further study of NAbs and the cells that produce them will further our understanding of the link between innate and adaptive immunity in turtles.

## 7. Mucosal Immunoglobulin

Mucosal immunity is an important arm of the innate immune defense. Polymeric immunoglobulin receptor (pIgR) is critical on mucosal surfaces to defend against invading pathogens through the transportation of polymeric Immunoglobulin (Ig) across epithelial tight junctions and the release of secretory Igs (Sigs) [161,162,163]. pIgR has been characterized in *P. sinensis* as closely related to avian and reptile pIgRs, with four Ig-like domains, and basal levels are upregulated in the oesophagus and intestines. *P. sinensis* pIgR expression was upregulated in vivo in response to LPS and *Aermonas sobria* challenge in the gastrointestinal tract [163]. IgA is the most abundant Ig isotype in the mammalian mucosal immune system. As turtles lack IgA, further characterization of the role of antibodies in mucosal immunity is required. For instance, *P. sinensis* strengthens mucosal immune functions in Winter when adaptive responses are lowered, with an increase in the expression of TLR2 and TLR4, and Muc2 by goblet cells [164]. Compensatory mechanisms are likely to exist in other species and understanding the impact of seasonality on mucosal antibodies across different species of turtle needs to be more fully explored.

## 8. Acute Phase Proteins

Acute phase proteins are used in wildlife diagnostics as biomarkers of immune response activity. The acute phase response is a systemic reaction of the host to local or systemic disturbances in homeostasis due to tissue injury, trauma, neoplastic growth, infection, or immunological disorders [165]. At the site of damage (infection or injury), pro-inflammatory cytokines are released from activated inflammatory cells, and the vascular system is activated. The activation of the innate immune effectors in immediate responses leads to the production of more cytokines and inflammatory mediators in blood and the extracellular fluid compartment. A systemic reaction is induced through activated cytokine receptors on various target cells, leading to the activation of the hypothalamic-pituitary-adrenal gland axis, reduction of growth hormones, and physical changes characterized by fever, anorexia, negative nitrogen balance, and catabolism of muscle cells [165]. The acute phase response can be measured in the blood through markers including higher levels of leukocytes, adrenocorticotrophic hormone (ACTH) and glucocorticoids, complement activation and blood coagulation, decreased levels of calcium, zinc, iron, vitamin A, and changes in concentration of plasma proteins [165].

Acute phase proteins have been measured in investigations for potential uses as biomarkers of immune response in *C. mydas* [73,74] and *C. caretta* [38,39]. Chaousis et al. employed a mass-spectrometry-based non-targeted proteomics approach and highlighted its potential for biomarker discovery in wildlife toxicology [73]. In this comparison study of green sea turtles from three different foraging sites in Australia, seven acute phase proteins were highly dysregulated; Alpha-1-antitrypsin, Alpha-2-macroglobulin-like, Ceruloplasmin, Complement C4, Complement factor, fibrinogen alpha chain, and haptoglobin [73]. Validation of simple assays to measure these proteins cheaply in small sample volumes would be an important step to understanding the impact of different environmental stressors and infections on the health of different populations of sea turtles. These assays would help to establish normal ranges for each protein incorporating change due to temperature and season. If adopted, the use of these markers in routine monitoring of sea turtles would be able to tell us important information about the impact of changes in their environment on turtle health.

## 9. Pattern Recognition Receptors (PPRs)

The innate immune system relies on pattern recognition receptors (PRRs) to identify conserved pathogen-associated molecular patterns (PAMPs), damaged senescent cells, and apoptotic host cells. PRRs distinguish these infectious microbial products and damage-associated molecular patterns (DAMPs) from healthy ‘self’ host cells [166,167]. Activated PRRs initiate two important immune responses; a cascade of signaling events to produce cytokines that begin nonspecific anti-infection and antitumor effects and begin the process for antigen-presenting cells (APCs) to activate antigen-specific adaptive immunity. There are five classes of PRRs: Toll-like receptors (TLRs), Retinoic-acid inducible gene I (RIG-I)-like receptors (RLRs), Nod-like receptors (NLRs), and C-type lectin receptors (CLRs), and cytosolic DNA sensors (cGAS, AIM2, IFI16) [167].

### 9.1. Toll-like Receptors

Toll-like receptors are type-1 transmembrane glycoproteins that are expressed either on the cell membrane or intracellular vesicles [168]. Ligation of TLRs triggers a downstream signaling cascade that activates the innate and then the adaptive immune response. The discovery of TLRs in *Drosophila melanogaster* was a landmark discovery for the understanding of innate immunity, and subsequently, TLRs were found to be highly conserved across vertebrates and some invertebrates [2,169,170]. Humans have 10 (TLR1-10) [171], fish have over 20 [172] and *C. mydas* has 11 TLRs [168]. The *C. mydas* genome contains TLR1, TLR2, TLR18, TLR3, TLR4, TLR5, TLR7, TLR8, TLR9, TLR21, and TLR22 genes [168]. Of these, mammals do not have TLR18, TLR21, or TLR22. However, these genes have been described in fish [173,174]. We will discuss this gene family in *C. mydas* and highlight duplicated genes or pseudogenes.

Gene duplication events can facilitate the evolution and expansion of a gene family’s function, specialized sub-functionalization, or differences in gene expression [175,176,177]. However, gene loss through mutations or pseudogenization can also be an adaptive evolutionary force depending on gene dispensability, mutational robustness, and environmental conditions [178]. The genomic diversity of TLRs in non-model species is a very interesting area of research, indicating evolution through both positive and negative selection because of infectious or environmental pressure. TLR members in different species can recognize different pathogenic ligands, thus differing in function. For example, human TLR5 only recognizes flagellin, while TLR5 in fish can recognize both bacterial and viral PAMPs, flagellin, and dsRNA, respectively [179]. TLRs are a result of positive selection and are constantly driven by evolutionary forces. More research is required to understand the specific patterns and pathways of innate defense that sea turtle PRRs have evolved.

TLR2 and TLR8 genes are duplicated pseudogenes in the *C. mydas* genome [168]. TLR2 is expressed in a variety of cells, including keratinocytes and sebaceous glands, which can produce bactericidal sebum to protect from skin infections. *C. mydas* and *P. sinensis* display a different number of TLR2 gene copies, perhaps highlighting the need for semi-aquatic turtles to have a more robust defense due to a thinner epidermis and increased exposure to pathogens [180,181]. TLR8 is an endosomal receptor that recognizes single-stranded RNA viruses, and this gene displays an interesting evolutionary pattern within *C. mydas*. Birds and squamate reptiles have no TLR8 ortholog, and mammals have one TLR8 gene. A duplication event (TLR8B) occurred in crocodilians and testudines, with a second duplication event occurring specific to testudines (TLR8C).

TLR8C is a pseudogene within *C. mydas* and a Mojave desert tortoise [177]. TLR8 gene duplication and loss events could represent a change in subspecialized function or an adaption to an abrupt environmental change.

We will simplify and discuss *C. mydas* TLRs in two groups; nonviral (TLR1, TLR2, TLR18, TLR4, and TLR5) and viral (TLR3, TLR7, TLR8, TLR9, TLR21, and TLR22). However, some TLRs can recognize both viral and bacterial-derived PAMPS and -induced DAMPS, respectively [168,182,183].

***Nonviral*.** TLR1, TLR2, and TLR18 are in the TLR1 subfamily in *C. mydas*. These TLRs form heterodimers to recognize triacylated lipoprotein and peptidoglycan from bacteria [183]. TLR1 and TLR2-dependent pro-inflammatory pathways are also upregulated by *C. caretta* in response to organic ultraviolet filters found in sunscreen agents [184]. TLR18 is widely expressed by aquatic animals. Fish increase TLR18 expression in response to challenges with *Aeromonas hydrophilia*, LPS, flagellin, and polyinosinic polycytidylic acid (poly(I:C)) [174,185,186]. The role of TLR18 in turtles is yet to be characterized.

TLR2 and TLR4 play a role in both the immune and reproductive systems of *P. sinensis*. TLR4 is the first discovered TLR and binds to MD-2 (myeloid differentiation factor 2), recognizing LPS molecules and components of *C. albicans*, *Trypanosoma*, and viruses [187]. TLR2, TLR4, and associated downstream target genes (MyD88, TNFα, IL-1β, and IL-6) were significantly upregulated in the *Pelodiscus sinensis* spleen. After the challenge with *Aeromonas hydrophilia* infection [66]. Turtle TLR expression is also regulated by estrogen in oviducts. Inflammatory pathways downstream of TLR2 and TLR4 were significantly downregulated in mated turtle oviducts in comparison to unmated ones. Estrogen receptor 1 (ESR1) and progesterone receptor (PGR) expression were upregulated, indicating an increase in hormone signaling can suppress the innate immune response to facilitate the long-term storage of spermatozoa [188]. As TLR2 and TLR4 are important PRRs for the initiation of the immune response to bacterial pathogens, the crosstalk between sex hormones and the regulation of pathogen recognition is an interesting and important area for future research.

***Viral***. TLR7, TLR8, and TLR9 comprise the TLR7 subfamily in *C. mydas*. TLR9 is found in mammals and in some chelonian genomes, including *C. mydas* and *P. sinensis*, but not in crocodilian or squamate [168]. TLR9, TLR21, and TLR22 recognize DNA with CpG motifs. However, reptilian TLR21 and TLR22 ligands have not been fully characterized [177].

Turtle TLRs have been investigated for their role in mediating the effect of diet on immune function. *Bacillis subtilis* B10 has been investigated as a pre-biotic to improve *P. sinensis* turtle health due to its immunomodulatory and antioxidant effects. Upregulation of the TLR8-dependent pathway and intestinal tight junction proteins in the gut and TLR5 in the liver was observed in animals on this diet [14]. This study suggests that dietary supplements and pre/probiotics that maintain gut microbial homeostasis can influence gut health and immunity in turtles. We are only at the beginning of understanding the diversity of the microbial communities that live within different niches of turtles’ gut/skin. We need to catalog these microbial communities and examine how they relate to health and disease. A major research priority is to understand how these microbes interact with each other and how these communities interact with their hosts.

The four other classes of PRRs, RLRs, NLRS, CLRs, and cytosolic receptors are lacking significant studies in reptiles, turtles included. TLRs can interact with RLRs and NLRs to coordinate an effective immune response. As evolutionary older organisms rely more heavily upon innate than adaptive responses [86,189], we speculate the PRRs in reptiles may have a more diverse function than in mammals in co-ordinating immune responses and other cellular functions of differentiation, proliferation, apoptosis, and autophagy.

### 9.2. RIG-I-like Receptors (RLRs)

RIG-I, melanoma differentiation-associated gene 5 (MDA5; also known as IFIH1), and laboratory of genetics and physiology 2 (LGP2) are RLRs. RIG-I and MDA5 detect cytosolic RNA, activate downstream signaling cascades resulting in phosphorylation of NF-κB and IRF3, and induce proinflammatory cytokines and interferons (IFN), respectively [190]. RIG-I is recognized for its diverse roles beyond innate immune activation, such as cellular proliferation and differentiation and cancer development [191]. RIG-I and MDA5 share similar structural homology but can respond differently to viruses. RLRs are conserved throughout reptile evolution, and MDA5 is reported to be under positive selection for the evolution of semi-aquatic reptiles [192]. The *C. mydas* genome contains LGP2, MDA5, and RIG-I [192].

There are few functional studies on reptilian RIG-I/MDA-5. However, one study of Chinese giant salamander RIG-I and MDA5 (*ad*RIG-I and *ad*MDA5) found that there was a high degree of homology with *C. mydas* genes as determined through BLAST analysis. RIG-I and MDA-5 gene expression was upregulated following iridovirus infection of Chinese giant salamanders [193]. Quantitative expression analysis of *ad*RIG-I and *ad*MDA5 revealed the highest expression in the spleen and lowest in the skin, and *ad*RIG-I is slightly higher expressed in muscle, heart, kidney, lung, and liver than *ad*MDA5 [193]. Reptile RLRs deserve further study to understand their role in the antiviral response.

### 9.3. Nod-like Receptors (NLRs)

NLRs are cytoplasmic receptors in immune cells (macrophages, dendritic cells, lymphocytes) and nonimmune cells (epithelial). NLRs are a highly expanded group with over 20 members in humans and 30 in mice [194]. NOD1 and NOD2 are very well described for their roles in peptidoglycan recognition and immune and apoptosis regulation. Notably, NOD2 is absent in all reptiles [192]. Reptile NLR gene structure and evolution have been described. However, functional characterization or role in infection has not been examined. *C. mydas* has differing functional domains within its NLR repertoire, NOD1, NLRX1, and NLRC3 when compared to other reptiles [195]. It is unknown what the significance of this is in terms of response to infection and immune regulation.

### 9.4. C-Type Lectin Receptors Are Primarily Expressed by Myeloid Cells

They mediate pathogen recognition by binding to specific microbial carbohydrates. This, in turn, mediates the inflammatory response and plays a key role in defense against viral and fungal infection [196]. There are no reports on turtle C-type lectin receptors. However, studies of bearded dragon, snake, and alligator genomes highlight the expanded repertoire of reptile CLR domains [197,198,199].

## 10. Interferons (IFN)

PRRs recognize their specific ligand and initiate a downstream signaling cascade resulting in the expression of proinflammatory cytokines, chemokinesand/or interferons (IFNs). IFNs are an essential part of the antiviral response, so named as they ‘interfere’ with viral replication [200]. IFNs induce the expression of a multitude of antiviral IFN-stimulated genes (ISGs) in a wide variety of cells and tissues and are critical for the induction of the antiviral response. Of the three subfamilies of IFN, IFN type I and II genes have been described in *P. sinensis*, while IFN type III has been described in green anole lizard (*Anolis carolinsis*) [11,201,202]. IFN-like activity has been observed in the kidney and peritoneal cells of tortoises in response to viral infection [203]. An IFN produced in response to the Saint Louis encephalitis virus was found in turtle heart cells that are chemically and physically similar to mammalian and avian IFN [204]. IFN-g gene has been described in *P*. *sinensis* [11].

IFN-g gene is constitutively expressed across most tissues [11]. IFN-γ gene expression was induced by LPS and Poly(I:C) in vitro, and transcription factors IRF1, IRF7, and STAT1 were upregulated. IFN-g stimulation induced ISGs, OAS, and PKR similar to the mammalian response [11]. IFN-g is produced by innate lymphoid cells, Natural killer (NK) cells, T lymphocytes, and plasmacytoid dendritic cells [205]. Type I IFN responses have not been described in *C. mydas*.

### Interferon Regulatory Factors (IRFs)

Interferon Regulatory Factors (IRF) are transcription factors that regulate the expression of type I IFN [183]. IRFs play a role in many immune responses, including apoptosis regulation, antimicrobial defense, and hematopoietic differentiation [200,206]. Eleven IRFs have been described in vertebrates, with 9 in humans and 6 in *C*. *mydas* (IRF1, IRF2, IRF4, IRF5, IRF6, and IRF8) [207]. IRF3, IRF7, and IRF9 are not described by Liu et al. These are important factors in producing type I IFN in mammalian cells and how turtles compensate for their absence needs to be elucidated [207].

## 11. Adaptive Immunity

The classic view of adaptive immunity is that T and B cells encode variable receptors that will bind to a specific antigen and instigate cytotoxic and memory responses. Reptiles’ adaptive response has been recently reviewed by Zimmerman [66]. Adaptive immunity involves antigen-specific immune responses and requires previous exposure to mount a full response. It is slower than the innate immune response, targets very specific antigens, and forms immunological memory to control re-infections [66] There is an important dialogue between innate and adaptive immune responses. The nature of the pathogen will dictate how it is sensed by the PRRs and the resulting innate immune response will stimulate either humoral or cell-mediated adaptive immunity.

## 12. Cell-Mediated Immunity

Cell-mediated immunity is predominantly dependent on T cell recognition and destruction of antigens, either by direct killing mechanisms or the secretion of cytokines to enhance the phagocytic ability of phagocytes [208]. T cells can recognize specific antigens of bacterial, viral, or parasitic origin or tumor cells [208]. T cell-mediated immunity can also result in autoimmune conditions due to aberrant recognition of self-antigens. The T-cell receptor (TCR), which contains CD3+ peptide chains, binds the specific antigenic peptide presented by the major histocompatibility complex (MHC) on APCs (antigen-presenting cells). The TCR/CD3 complex is found in all jawed vertebrates [209]. Dependent upon cytokines in the cellular environment, T-helper cells (CD4+) differentiate into five T-cell subtypes (Th1, Th2, Th17, Treg, or Th9) and produce cytokines to stimulate T-cell effector functions and B-cell antibody production or form memory T-cells. Cytotoxic T cells (CD8+) release cytotoxic granules upon antigen recognition that kill virally infected or tumor cells [208].

CD3+ T lymphocytes have been identified in the blood and tissue of *C. mydas* [209]. In this study, the authors found that an antibody raised against human CD3 cross-reacted with CD3 expressed by turtle T cells. This antibody recognized a conserved region of the epsilon chain of CD3. These are assumed to be helper and cytotoxic T cells. However, we do not yet understand if these cells acquire different effector functions that align with murine or human T cell subsets [66,67]. The study of cellular immunology in non-model organisms is limited by a dearth of reagents that recognize and discriminate different cell types. Understanding how key molecules are conserved across species will allow the identification of further antibodies that can be used to study the function of T cells in species such as turtles. For example, the genetic components for CD4+ and CD8+ T-cells are described in the western painted turtle (*Chrysemys p. bellii*) and giant tortoises (*Chelonoidis abingdonii*, (Quoy and Gaimard, 1824) and *Aldabrachelys gigantea* (Schweigger, 1812)) [32]. Further comparative analysis of these T cell receptor/T cell signaling pathways across reptiles and mammals will further our understanding of how the cellular response is regulated in turtles and potentially increase access to tools to undertake functional studies.

Lymphocyte proliferation is measured as a biomarker for turtle adaptive immune health. Concanavalin A (ConA) and phytohemagglutinin (PHA) have been examined as T-cell mitogens and LPS as a B-cell mitogen. Turtles have a decreased T and B cell mitogen response in comparison to mammals [50]. T cell-mediated immunity is compromised in turtles with severe FP. Thus, immunosuppression is a marker of late-stage disease [88,210]. In FP-afflicted turtles, both T and B cell proliferation in response to mitogens was reduced in turtles with moderate and severe FP [49,50,53,88]. In early-stage tumors, there is a measurable immune response, an increase of CD3+ lymphocytes, and upregulation of key immune pathways (leukocyte and lymphocyte pathways). In late-stage tumors, there were fewer CD3+ T-cells with downregulation of immune and apoptotic genes [211].

T cell lymphocyte proliferation is affected by environmental contaminants such as organochlorides and mercury in *C. caretta* [36,135]. The effects of heavy metal accumulation (Cd, Cr, Cu, Pb, and Hg) on red-eared slider turtle lymphocyte proliferation were examined with a PHA skin test. A PHA skin test stimulates T-cell proliferation, differentiation, and cytokine production with inflammation and swelling at the injection site. The accumulation of lead (Pb) in kidneys may be immunosuppressive and disrupt T-lymphocyte proliferation [212].

In mammals, regulatory T cells (Tregs) are responsible for immune suppression to avoid overreaction of the immune system causing damage to an organism’s tissues [67,208]. The transcription factor FoxP3 is required for the differentiation of Treg precursors and needs further study in reptiles. It is thought that if FoxP3 is not functional in reptiles, however the TGF-β cytokine may play a role in inducing T-cell tolerance [67]. As *C. mydas* lack germinal centers and lymph nodes, T cells may play a different role in immunity than mammals [115]. T cell proliferation is seasonally dependent, as the thymus and spleen change during the seasons [86]. Turtles are likely to have unique mechanisms underpinning T-cell proliferation, differentiation, and memory responses. As we develop better methods to assess cellular immunity in non-model species, we will gain insight into the evolution of adaptive immunity.

## 13. Humoral Immunity

The antibodies produced by B cells are the backbone of the humoral immune response. Antibodies can neutralize toxins, prevent pathogen internalization, and opsonize targets for phagocytosis. B cells are activated and differentiate into antibody-secreting plasma cells in the presence of an antigen and are assisted by helper CD4+ T cells [213]. In mammals, there are two subtypes of B cells, the evolutionarily older B-1 subset, which produces natural antibodies (Nabs) before and after antigen stimulation, and the B-2 subset, which produces highly variable and specific antibodies after stimulation by an antigen. B-1 cells in turtles are similar to fish B-1 cells, and evidence for the B-2 cell subset is lacking, and it is suggested that turtles do not have them [66,110]. However, a memory response is induced, and the role of B cells in reptiles deserves further study.

Turtle B-1-like cells with ‘like’ functions of phagocytosis, lack of senescence, and production of non-specific, polyreactive antibodies have been described in red-eared slider turtle (*T. scripta*) [20] and Mojave desert [158] and Gopher tortoises [214] (*Gopherus agassizii* (Cooper, 1861) and *Gopherus polyphemus* (Daudin, 1801)) [66]. As mammalian B cells are found in secondary lymphoid structures not found in reptiles, such as lymph nodes and germinal centers formed by activated B-2 cells (previously discussed, see Figure 2), it is suggested that reptilian B cells are not limited to anatomical locations such as peritoneal and pleural cavities [66].

Phagocytic B cells in fish have microbicidal activity and can function as antigen-presenting cells to stimulate the adaptive immune response [215]. B lymphocytes in *C. mydas* have phagocytic activity, and this ability is not affected by FP [51,111]. B cell characteristics described in fish that are lacking studies in turtles include lack of proliferation upon antigen stimulation of the B-cell receptor (BCR), expression of B-1 specific cell markers and innate receptors, and activation in T-dependent and T-independent manner [66,216,217]. Further study is required on the function of phagocytic B cells in both *C. mydas* and all reptiles [66,67]. Furthermore, the regulatory role of B cells is not well understood. A study of desert tortoises suggests that B cells play a role in polarizing the immune system to avoid a dangerous inflammatory response [158]. This study suggests that B cells play a role in controlling inflammation.

Immunoglobulin (Ig) molecules are composed of one heavy (H) and one light (L) protein chain joined by disulfide bonds [213]. Mammals have five classes of Ig (M, D, G, E, and A) based on their heavy chain isotypes, and each class displays a differentiated function (complement fixation, unclear, opsonization, parasite infection, mucosal immunity, respectively). Diversity in Ig isotypes represents an evolution in immune function [218]. IgE and IgG in mammals are derived from IgY genes [219].

IgM, IgD, and IgY genes have been described in Chinese soft-shelled turtles, pond slider turtles, and red-eared sliders (*P. sinensis*, *Pseudemys scripta*, and *Trachemys scripta elegans*) [25,220,221,222,223]. The genomic structure of four turtle Ig classes has been described in *P. sinensis* and *Chrysemys picta* belli, with one gene each for IgD and IgM and multiple genes for IgY and IgD2-like) [223]. *P. sinensis* and *C. picta* have 17 and 13 sequences, respectively, coding for Ig isotypes [223]. *C. mydas* has 30 immunoglobulin genes, with one gene each for IgM and IgD and 14 genes each for IgD2 and IgY [218].

*C. mydas* have three described Ig structures [58,60]. Turtle Igs include a presumed pentamer 17S IgM and two monomeric IgYs: 7S IgY and 5.7S IgY. The 5.7S IgY is composed of two equally sized heavy and light chains with a truncated Fc portion, which is also found in ducks [58,224,225]. The 7S IgY has similar functions to human IgG, contains two differently sized heavy chains bound to light chains, and is presumed noncovalently associated with a 90-kDa moiety that is secreted during chronic antigenic stimulation [58]. Unlike mammals, reptile IgY is thought to lack a hinge region due to a lack of molecular evidence for one. However, the flexibility of the Fab arms is maintained [58]. Work et al. reported *C. mydas* to mount a response to glycoprotein antigens of ChHV5 using 7S and 5.7S IgY antibodies [226].

Monoclonal antibodies for the detection of turtle immunoglobulins via competitive ELISA have been developed for *C. caretta* [40] and *C. mydas*. Reference intervals have been established for Ig in *C. mydas* (0.40–0.85 g/dL) and *C. carreta* (0.38–0.94 g/dL) [227]. These assays can be used to monitor antibody responses in turtles and further refined to measure antigen-specific antibody titers. This would allow us to gain further insight into the importance of antibodies in the immune response in turtles.

## 14. Conclusions

In conclusion, sea turtles’ immune system is a robust defense capable of protecting the host from microorganisms or cancer. However, they relied upon innate response is highly sensitive to natural and anthropogenic environmental conditions. The lack of reliance upon the adaptive immune response is less understood. Characterization or confirmation of T and B cell subsets would elucidate the importance of the adaptive response. PRRs and the interferon response have more than one function of monitoring for PAMPS or DAMPS. They exert control over cellular processes like apoptosis, proliferation, and differentiation. The antiviral defense is less understood, and knowledge of interferons, their regulatory functions, and ISG expression could be employed to assist cancer treatments. Turtles’ antiviral and anticancer immune functions are yet to be fully characterized.

Sea turtles are a sentinel species, as they are long-lived obligatory inhabitants of near-shore ecosystems [228,229,230,231,232]. The health of a sentinel species is monitored as a measure of the health of the marine environment [230,233]. Sea turtle immune function is sensitive to fluctuations in temperature [87] and to pollutants in the environment, such as brevotoxins [22,37,52,134] and chemical disruptions [24,31]. Rapid changes in the marine environment are thought to make marine life more vulnerable to diseases and infections. However, this exact mechanism of action is poorly understood [230,231].

Earth is on track to enter a sixth mass extinction event as the current loss of biodiversity caused by anthropogenic actions is at a quicker rate than any other extinction event caused by natural phenomena [234,235]. Testudines are particularly vulnerable, and it is estimated that if the current rate of extinction continues that massive losses of turtle biodiversity will occur in this lifetime, and the entire group could disappear in a few short centuries [236,237]. Turtles have survived at least two mass extinction events and have proven adaptability to new extreme environments [237,238]. Ecological opportunities such as recovery from mass extinction, invasion of new habitats, loss of predators, or development of key phenotypic traits can trigger periods of adaptive radiations in which fast evolutionary rates facilitate expansion and loss of clades. However, this occurs on timescales encompassing millions of years [234,239,240]. Testudines have the potential to recover from modern extinction rates through reducing the risk of disease to current species and populations. Increased knowledge of the sea, freshwater, and terrestrial turtle immune systems will aid this aim, provide insight into the ancient evolution of host defense against pathogens, and potentially inform future precision medicine therapies.

## Figures and Tables

**Figure 1 animals-13-00556-f001:**
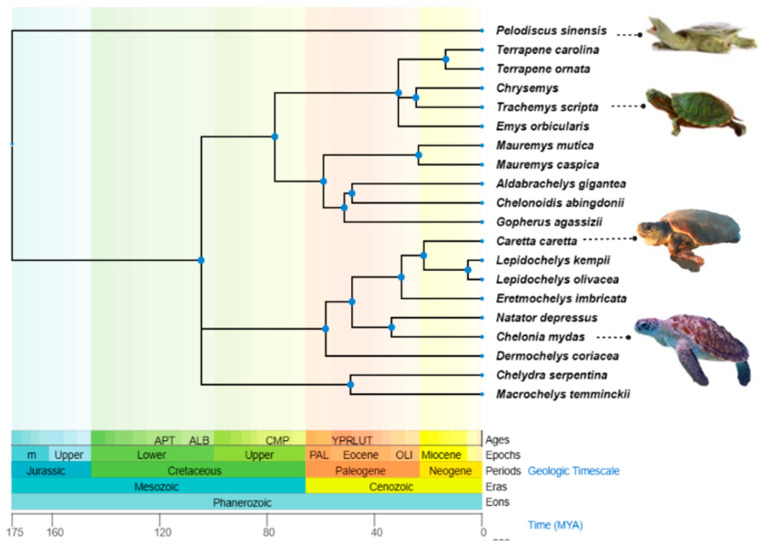
The phylogenetic tree of Testudines species is mentioned in this review. The tree was constructed from the information provided by http://timetree.org/ (accessed on 20 December 2022).

**Figure 2 animals-13-00556-f002:**
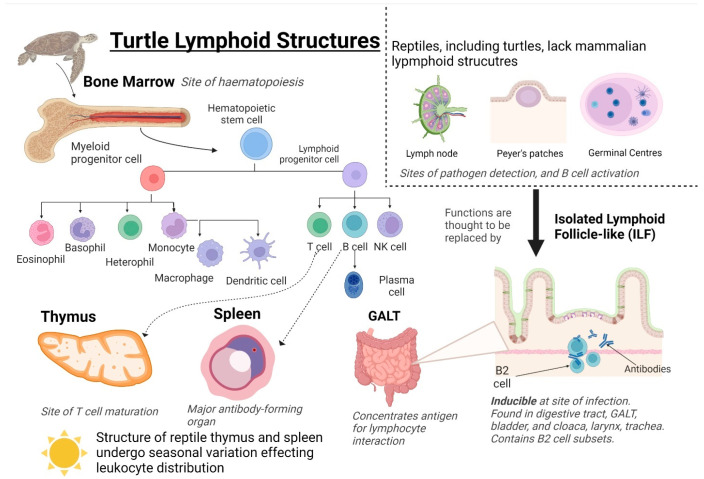
**Turtle Lymphoid Structures.** Primary structures include the bone marrow and thymus, and secondary structures include the spleen, Gut-associated lymphoid tissue (GALT), and isolated lymphoid follicle-like (ILFs). Hematopoietic stem cells are generated from the bone marrow and differentiate into myeloid progenitor cells or lymphoid progenitor cells. Myeloid progenitor cells differentiate into innate effector cells; granulocytes: basophils and eosinophils, and phagocytes: heterophils, monocytes, and macrophages. Lymphoid progenitor cells differentiate into adaptive immune effector cells: T and B cells. The distribution and differentiation of leukocytes are affected by seasonal variations in the structure of the spleen and thymus. Created using BioRender.

**Figure 3 animals-13-00556-f003:**
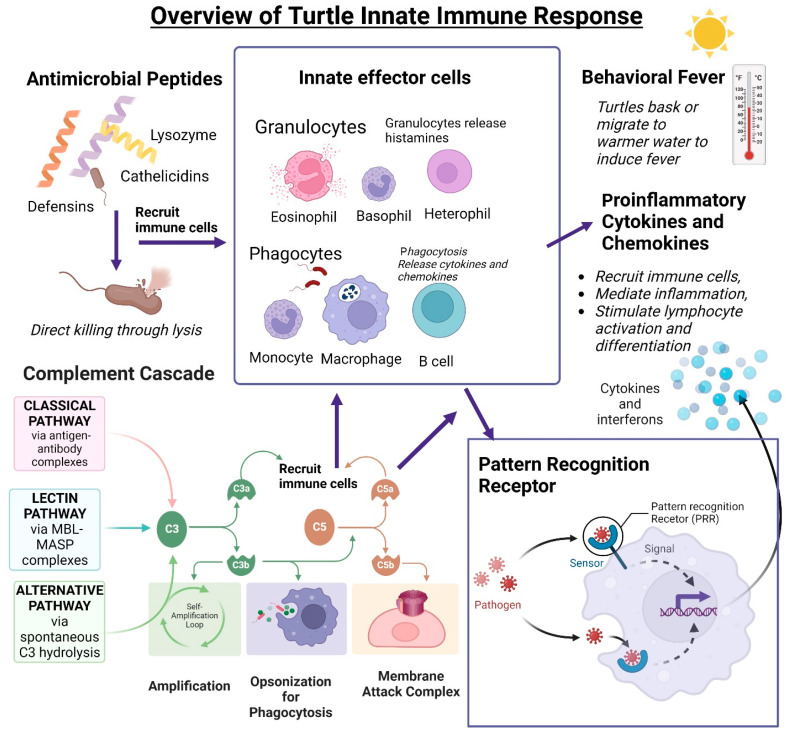
Overview of turtle innate immune functions. Nonspecific innate responses require no previous exposure to microorganisms to ensure a robust response. Antimicrobial peptides (lysozymes, cathelicidins, and defensins) can directly kill microorganisms by cell wall permeabilization and induction of lysis. Innate effector cells include granulocytes and phagocytes. Granulocytes are heterophils, basophils, and eosinophils and can release histamines when activated by microorganisms. Reptiles, including *C. mydas*, have phagocytic B cells. The complement cascade activates C3 through the classical, lectin, or alternative pathways, which results in the opsonization of microorganisms for phagocytosis or lyse cell wall through the membrane attack complex (MAC) formation. Cleaved C3a and C5a recruit nonspecific effector cells. Innate effector cells recognize microorganisms through the binding of pattern recognition receptors which begins a downstream signaling cascade ending in the release of cytokines and interferons. Cytokines and chemokines can recruit immune cells and mediate inflammation and lymphocyte activation and differentiation. Created using BioRender.

**Table 1 animals-13-00556-t001:** Overview of Testudines habitat, ecosystem, and immune system studies reviewed.

Testudines	GenBank Common Name	Habitat	Ecosystem	Refs.
*Pelodiscus sinensis*	Chinese soft-shelled turtle	East Asia	Aquatic	[9,10,11,12,13,14,15,16]
*Terrapene carolina*	Eastern box turtle	North America	Terrestrial	[17,18]
*Terrapene ornata*	Ornate box turtle	North America	Terrestrial	[17]
*Chrysemys picta bellii*	Western painted turtle	North America	Aquatic	[18]
*Trachemys scripta*	Red-eared slider turtle	North America	Semi-aquatic	[19,20,21,22,23,24,25]
*Emys orbicularis*	European pond turtle	Europe, West Asia, North America	Aquatic	[26]
*Mauremys mutica*	Yellowpond turtle	East Asia	Aquatic	[27]
*Mauremys caspica*	Caspian turtle/striped-necked terrapin	Eastern Mediterranean	Aquatic	[26,27,28,29,30,31]
*Aldabrachelys gigantea*	Aldabra giant tortoise	Seychelles	Terrestrial	[32]
*Chelonoidis abingdonii*	Abingdon island giant tortoise	Galápagos	Terrestrial	[32]
*Gopherus agassizii*	Agassiz’s desert tortoise	North America	Terrestrial	[33]
*Caretta caretta*	Loggerhead turtle	AO, PO, IO, and MS	Marine	[34,35,36,37,38,39,40,41]
*Lepidochelys kempii*	Kemp’s ridley sea turtle	AO	Marine	[42,43,44]
*Lepidochelys olivacea*	Olive ridley sea turtle	PO	Marine	[45,46,47]
*Eretmochelys imbricata*	Hawksbill sea turtle	IP and AO	Marine	[48]
*Natator depressus*	Flatback sea turtle	PO	Marine	*
*Chelonia mydas*	Green sea turtle	AO, PO, IO, and MS	Marine	[49,50,51,52,53,54,55,56,57,58,59,60]
*Dermochelys coriacea*	Leatherback sea turtle	AO, PO, and IO	Marine	*
*Chelydra serpentina*	Common snapping turtle	North America	Aquatic	[61,62,63,64]
*Macrochelys temminckii*	Alligator snapping turtle	North America	Aquatic	[61]

List of Testudines’ immune response reviewed, regarding species name, habitat, and ecosystem. * Sea turtles lack significant studies on the immune response. Atlantic Ocean (AO), Pacific Ocean (PO), Indian Ocean (IO), and Mediterranean Sea (MS).

## Data Availability

Not applicable.

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
