# Peer review of "Immunity in Sea Turtles: Review of a Host-Pathogen Arms Race Millions of Years in the Running"

_animals, 2023, doi:10.3390/ani13040556_

Round 1

Reviewer 1 Report

it needs major revision and reorientation 

Author Response

Reviewer 1

  1. ‘Title: it seems very rudimentary it needs to reorientation as per the content. I would suggest it should be ended with question marks’.

We have thought about this comment and have revised the title of the manuscript as follows: Immunity in Sea Turtles: results a review of a host-pathogen arms race millions of years in the running. 

This better describes the content; it is a broad subject and the title of the article needs to reflect this and the impact of longevity on the evolution immunity in sea turtles.

  1. ‘Abstract: line 27-28, it seems bit contradictory as coastal waters are open water where pollution and other terms like diseases are still not clear. So, it need reconfirmation.’

Line 35: We accept the correction and replaced the word coastal with marine.

  1. ‘Introduction Author should provide the reference which are recent in the same field. Since, it is a review article so I suggest to improve the language and other phrasing mistakes. I could not find any table in whole MS so I definitely suggest to improve the text by putting the table in concerned section with latest findings. Its highly mandatory.’

While adhering to reviewer one and reviewer two’s helpful comments, we have re-written the introduction and will discuss the extensive revision subsequently (Reviewer 2). We have addressed Reviewer One comments to enhance the Introduction section:

Line 59: We have added Table 1 with select references regarding the immune system of Testudines reviewed within the manuscript; ‘The geographic distribution and phylogenetic relationships of the turtle species discussed in this review are outlined in Table 1 and Figure 1’.

Line 106- 111: Table 1 (page 6): Overview of Testudines habitat, ecosystem and immune system studies reviewed’.

  1. ‘The while MS need reorientation for the content it seems a flaws presentation.’

We have carefully edited the manuscript to ensure that the formatting, grammar, and presentation adhere to the journal guidelines.

  1. Author should provide a distribution GIS map for showing the habitats for turtle.
  2. Classification of the turtles

In response to reviewer one comments 3, 5, and 6, we have added a new Figure 1, which supports the Table 1 information. The new table and figure highlight the phylogenetic relationships of Testudines mentioned within review, and classification according to their scientific names, habitats, ecosystems, with select references on the immune system for each species available.

Line 112- 115: Figure 1 (Page 7): Phylogenetic relationship of Testudines reviewed.

Reviewer 2 Report

Immunity in Sea Turtles: Results of a Host-Pathogen Arms 2 Race Millions of Years in the Running by Alana Nash 1 and Elizabeth J. Ryan. animals-2101845animals-2101845

Overview:  This is an impressive effort, and clearly the authors have done a ton of work assembling this material.  It does a good job bringing together existing literature on the topic.  My major critique is that the entire MS reads a bit like a book report and not a true critical review.  In their revision, authors may want to ponder the following:

1) What are the major gaps that exist?

2) What might be suggested approaches to address those gaps?

Additional comments are within MS as text or edits.

Finally consider the following articles to cite:

https://doi.org/10.1016/j.vetimm.2013.09.004

 https://doi.org/10.1638/2010-0228R4.1

Author Response

Reviewer 2

We wish to express our sincere gratitude to Reviewer 2’s insightful comments and we list below the improvements made to the manuscript under guidance from their suggested references and ideas.

 1) What are the major gaps that exist?

2) What might be suggested approaches to address those gaps?

3) is this review important or timely?  Other than summarizing literature, how does your review help chart a path forward for the field of green turtle immunology?  

We addressed these helpful comments through re-writing of introduction. We have also carefully revised the whole document and incorporated additional comments to ensure we use this opportunity to chart a path forward in this research area.

  • Line 69-75: Major gaps identified.

The anti-viral response in particular is poorly understood [10–12]. Turtles’ lymphocytes are thought to respond to infection in a relatively nonspecific way with an altered antibody and memory response [10]. Large gaps in knowledge of the sea turtle immune system include; the mechanism of pathogen recognition through pattern recognition receptors (PRRs), the pathways and efficiency of antiviral effectors such as interferons (IFNs) and IFN-stimulated genes (ISGs), the role of cytotoxic T cells in killing infected or damaged host cells, and the mechanism and locations of T cell-mediated B cell activation [10].

  • Line 87-89: Suggested approaches to address gaps identified.

Multidisciplinary interest in sea turtle immunology is growing, guided by ecoimmunology and ‘One Health’ principles, and application of omics techniques, protein isolation and purification, microscopy and histology will aid in characterisation of immune components [9,12,31]. The adaption, standardization, and application of these techniques to sea turtles will continue to identify dysregulated and disease related pathways, allowing the selection of potential precision medicine treatments [15,31].

  • Lines 76- 86: Why the review is important and timely is addressed.

…For example, fibropapillomatosis (FP) is a tumorous disease that affects sea turtles. The dramatic increase in FP prevalence since the 1980s is attributed to a virus and an unknown environmental factor, thought to be anthropogenic in nature [13–15]. Results from genomic [16], proteomic [17–19] and transcriptomic studies [15,20,21] have consistently identified immune genes as dysregulated between healthy and diseased turtles, or by levels of exposure to marine contaminants [18]. Additionally, sea turtles are possible reservoirs for zoonotic viruses as C. mydas, along with snakes and pangolins, have been suggested as potential intermediate hosts of SARS-CoV-2 [22]. Incidences of antibiotic-resistant bacterial infections in sea turtles have been increasing as antibiotics are continually released into the environment [23–30].

Comments within MS:

  • Line 103: We inserted commas and references for clarity: Furthermore the mydas genome was the first published, and thus most investigated, sea turtle genome (GenBank: GCA_000344595.1) [60].
  • Line 182: We added suggested reference and improved clarity in the following sentence: This antimicrobial epidermal barrier is surveilled by Langerhans+ cells and is aided by antimicrobial peptides (AMPs) [102,103].
  • Line 185: We corrected the manuscript and added ‘behavioral fever’.
  • Line 238: Spelling of recognition corrected in diagram.
  • Line 250: Spelling of BioRender corrected (*correct in OG manuscript)
  • Line 205: We listed the turtle phagocytic cells including monocytes.
  • Line 211-213: We have edited, clarified information, and added suggested references.
  • Line 252: We accepted suggestion: Fever is an integral part of the innate immune response in homeotherms.
  • Line 262-269: We added the suggested reference, and we expanded on the relationship between fever and cancer in murine models:

Certain C. mydas populations in the Pacific Ocean, including Hawaii, engage in terrestrial basking behaviour, due to cold winter sea surface temperature 123. A 2006 study observed a relationship between FP and basking behavior in C. mydas. Only turtles with FP were observed basking, resulting in a 2.9°C increase above ambient body temperature 124. Heat-seeking behavior has been reported in murine cancer models, and cold stress is associated with accelerated tumour growth 125,126. Further study is needed to define the relationship between infection and mechanisms governing behavioral fever in sea turtles, and to understand if warming oceans will impact on the fever response, and how this will affect immune response to infection and incidence of cancer.

  • Line 603- 609: We condensed and clarified paragraphs and we added suggested references.

T cell-mediated immunity is compromised in turtles with severe FP; thus immunosuppression is a marker of late-stage disease [37,212]. In FP-afflicted turtles, both T and B cell proliferation in response to mitogens was reduced in turtles with moderate and severe FP [36–38,44].

  • Line 640: We have added discussion and cited the suggested reference.

B-1 cells in turtles are similar to fish B-1 cells, and evidence for B-2 cell subset is lacking, and it is suggested that turtles do not have them [10,111]. However, an anamnestic response to secondary immunization is reported in turtles, including C. mydas 36. The production of antigen-specific antibodies by B-cell subsets, through T-cell-mediated B-cell activation, and subsequent role in antiviral or antitumour defence in turtles deserves further consideration to aid potential therapy development.

  • Line 640: We corrected the type mistake. Added ‘FP’. ‘B lymphocytes in C. mydas have phagocytic activity, this ability is not affected by FP’.
  • Line 650: We added the reference: IgE and IgG in mammals are derived from IgY genes [220].
  • Line 664: We added the helpful suggested reference:
  1. mydas and C. caretta mount a response to glycoprotein antigens of ChHV5 using 7S and 5.7S IgY antibodies, with varying degrees of response to FP presence or severity [229,230].
  • Line 671: We cited suggested references:

Monoclonal antibodies for the detection of turtle immunoglobulins via competitive ELISA have been developed for C. caretta [78] and C. mydas [37,226].

  • Line 674- 687 (previous MS): We accepted the suggestion to delete paragraphs to enhance clarity of the manuscript.

Reviewer 3 Report

This work is a solid review for researchers investigating immune response of sea-turtles. 

However, there are minor mistakes which must be corrected. 

1) The authors sometimes do not follow the rules of the journal and the citation is wrongly presented - i.e. the author/authors and year is provided in the text instead of the number.

2) The generic and species names are not always in italic as they should be according to ICZN.

3) When using a species name for the first time in the manuscript is should always have a name of a person who discovered it e.g. Chelonia mydas (Linnaeus, 1758). You can easily check it on reptile-databse web page.

4) Check also text, there are some minor mistakes such as: different font in the text, double spaces and so on.

I provided comments directly in the body of the manuscript.

Author Response

Reviewer 3

  1. The authors sometimes do not follow the rules of the journal and the citation is wrongly presented - i.e. the author/authors and year is provided in the text instead of the number.
  2. The generic and species names are not always in italic as they should be according to ICZN.
  3. When using a species name for the first time in the manuscript is should always have a name of a person who discovered it e.g. Chelonia mydas (Linnaeus, 1758). You can easily check it on reptile-database web page.

We have ensured the citations are in number format, and generic and species names are in italics and accompanied by the appropriate name as according to the ICZN, while adhering to the rules of the journal.

  1. Check also text, there are some minor mistakes such as: different font in the text, double spaces and so on.

We list below the necessary changes made to reflect the helpful comments made by Reviewer 3 in the manuscript:

  • Line 69 (previous MS)/ Line 84 (revised MS): Corrected font.
  • Line 79 / 102: Corrected double space.
  • Line 130 / 151: Corrected full stop
  • Line 135- 149/ 155- 167: Corrected double space
  • Line 390/ 407: double space
  • Line 424/ 441: font corrected.
  • Line 459, 462/ 422, 423: italic
  • Line 464 / Line 426: references correct format
  • Line 482/ Line 442: corrected italic names
  • Line 493 / Line 452: italic names for C. mydas and P. sinensis. (note italics name format was included in the original MS. Maybe a double check for typesetters?)
  • Line 577 / Line 527: space.
  • Line 684: corrected reference format

Round 2

Reviewer 1 Report

Authors has included all comments so it may be processed further for publication.